# MaCow: Masked Convolutional Generative Flow

**Xuezhe Ma, Xiang Kong, Shanghang Zhang, Eduard Hovy**
Carnegie Mellon University
Pittsburgh, PA, USA
`xuezhem,xiangk@cs.cmu.edu, shanghaz@andrew.cmu.edu, hovy@cmu.edu`

## Abstract

Flow-based generative models, conceptually attractive due to tractability of the exact log-likelihood computation and latent-variable inference as well as efficiency in training and sampling, has led to a number of impressive empirical successes and spawned many advanced variants and theoretical investigations. Despite computational efficiency, the density estimation performance of flow-based generative models significantly falls behind those of state-of-the-art autoregressive models. In this work, we introduce *masked convolutional generative flow* (**MaCow**), a simple yet effective architecture for generative flow using masked convolution. By restricting the local connectivity to a small kernel, MaCow features fast and stable training along with efficient sampling while achieving significant improvements over Glow for density estimation on standard image benchmarks, considerably narrowing the gap with autoregressive models.

## 1 Introduction

Unsupervised learning of probabilistic models is a central yet challenging problem. Deep generative models have shown promising results in modeling complex distributions such as natural images (Radford et al., 2015), audio (Van Den Oord et al., 2016) and text (Bowman et al., 2015). Multiple approaches emerged in recent years, including Variational Autoencoders (VAEs) (Kingma and Welling, 2014), Generative Adversarial Networks (GANs) (Goodfellow et al., 2014), autoregressive neural networks (Larochelle and Murray, 2011; Oord et al., 2016), and flow-based generative models (Dinh et al., 2014, 2016; Kingma and Dhariwal, 2018). Among these, flow-based generative models gained popularity for this capability of estimating densities of complex distributions, efficiently generating high-fidelity syntheses, and automatically learning useful latent spaces.

Flow-based generative models typically warp a simple distribution into a complex one by mapping points from the simple distribution to the complex data distribution through a chain of invertible transformations with Jacobian determinants that are efficient to compute. This design guarantees that the density of the transformed distribution can be analytically estimated, making maximum likelihood learning feasible. Flow-based generative models have spawned significant interests for improving and analyzing its algorithms both theoretically and practically, and applying them to a wide range of tasks and domains.

In their pioneering work, Dinh et al. (2014) first proposed *Non-linear Independent Component Estimation* (NICE) to apply flow-based models for modeling complex high-dimensional densities. RealNVP (Dinh et al., 2016) extended NICE with a more flexible invertible transformation to experiment with natural images. However, these flow-based generative models resulted in worse density estimation performance compared to state-of-the-art autoregressive models, and are incapable of realistic synthesis of large images compared to GANs (Karras et al., 2018; Brock et al., 2019). Recently, Kingma and Dhariwal (2018) proposed Glow as a generative flow with invertible $1 \times 1$ convolutions, which significantly improved the density estimation performance on natural images. Importantly, they demonstrated that flow-based generative models optimized towards the plain

likelihood-based objective are capable of generating realistic high-resolution natural images efficiently. Prenger et al. (2018) investigated applying flow-based generative models to speech synthesis by combining Glow with WaveNet (Van Den Oord et al., 2016). Ziegler and Rush (2019) adopted variational inference to apply generative flows to discrete sequential data. Unfortunately, the density estimation performance of Glow on natural images remains behind autoregressive models, such as PixelRNN/CNN (Oord et al., 2016; Salimans et al., 2017), Image Transformer (Parmar et al., 2018), PixelSNAIL (Chen et al., 2017) and SPN (Menick and Kalchbrenner, 2019). There is also some work (Rezende and Mohamed, 2015; Kingma et al., 2016; Zheng et al., 2017) trying to apply flow to variational inference.

In this paper, we propose a novel architecture of generative flow, *masked convolutional generative flow* (**MACOW**), which leverages masked convolutional neural networks (Oord et al., 2016). The bijective mapping between input and output variables is easily established while the computation of the determinant of the Jacobian remians efficient. Compared to inverse autoregressive flow (IAF) (Kingma et al., 2016), MACOW offers stable training and efficient inference and synthesis by restricting the local connectivity in a small "masked" kernel as well as large receptive fields by stacking multiple layers of convolutional flows and using rotational ordering masks (§3.1). We also propose a fine-grained version of the multi-scale architecture adopted in previous flow-based generative models to further improve the performance (§3.2). Experimenting with four benchmark datasets for images, CIFAR-10, ImageNet, LSUN, and CelebA-HQ, we demonstrate the effectiveness of MACOW as a density estimator by consistently achieving significant improvements over Glow on all the three datasets. When equipped with the variational dequantization mechanism (Ho et al., 2019), MACOW considerably narrows the gap of the density estimation with autoregressive models (§4).

## 2 Flow-based Generative Models

In this section, we first setup notations, describe flow-based generative models, and review Glow (Kingma and Dhariwal, 2018) as it is the foundation for MACOW.

### 2.1 Notations

Throughout the paper, uppercase letters represent random variables and lowercase letters for realizations of their corresponding random variables. Let $X \in \mathcal{X}$ be the random variables of the observed data, e.g., $X$ is an image or a sentence for image and text generation, respectively.

Let $P$ denote the true distribution of the data, i.e., $X \sim P$, and $D = \{x_1, \ldots, x_N\}$ be our training sample, where $x_i, i = 1, \ldots, N$, are usually i.i.d. samples of $X$. Let $\mathcal{P} = \{P_\theta : \theta \in \Theta\}$ denote a parametric statistical model indexed by the parameter $\theta \in \Theta$, where $\Theta$ is the parameter space. $p$ denotes the density of the corresponding distribution $P$. In the deep generative model literature, deep neural networks are the most widely used parametric models. The goal of generative models is to learn the parameter $\theta$ such that $P_\theta$ can best approximate the true distribution $P$. In the context of maximum likelihood estimation, we minimize the negative log-likelihood of the parameters with:

$$\min_{\theta \in \Theta} \frac{1}{N} \sum_{i=1}^{N} -\log p_\theta(x_i) = \min_{\theta \in \Theta} \mathrm{E}_{\widetilde{P}(X)}[-\log p_\theta(X)], \tag{1}$$

where $\tilde{P}(X)$ is the empirical distribution derived from training data $D$.

### 2.2 Flow-based Models

In the framework of flow-based generative models, a set of latent variables $Z \in \mathcal{Z}$ are introduced with a prior distribution $p_Z(z)$, which is typically a simple distribution like a multivariate Gaussian. For a bijection function $f : \mathcal{X} \rightarrow \mathcal{Z}$ (with $g = f^{-1}$), the change of the variable formula defines the model distribution on $X$ by

$$p_\theta(x) = p_Z\left(f_\theta(x)\right) \left| \det\left(\frac{\partial f_\theta(x)}{\partial x}\right) \right|, \tag{2}$$

where $\frac{\partial f_\theta(x)}{\partial x}$ is the Jacobian of $f_\theta$ at $x$.

The generative process is defined straightforwardly as the following:

$$\begin{aligned} z &\sim p_Z(z) \\ x &= g_\theta(z). \end{aligned} \tag{3}$$

Flow-based generative models focus on certain types of transformations $f_\theta$ that allow the inverse functions $g_\theta$ and Jacobian determinants to be tractable to compute. By stacking multiple such invertible transformations in a sequence, which is also called a (normalizing) *flow* (Rezende and Mohamed, 2015), the flow is then capable of warping a simple distribution ($p_Z(z)$) into a complex one ($p(x)$) through:

$$X \underset{g_1}{\overset{f_1}{\longleftrightarrow}} H_1 \underset{g_2}{\overset{f_2}{\longleftrightarrow}} H2 \underset{g_3}{\overset{f_3}{\longleftrightarrow}} \cdots \underset{g_K}{\overset{f_K}{\longleftrightarrow}} Z,$$

where $f = f_1 \circ f_2 \circ \cdots \circ f_K$ is a flow of $K$ transformations. For brevity, we omit the parameter $\theta$ from $f_\theta$ and $g_\theta$.

## 2.3  Glow

Recently, several types of invertible transformations emerged to enhance the expressiveness of flows, among which Glow (Kingma and Dhariwal, 2018) has stood out for its simplicity and effectiveness on both density estimation and high-fidelity synthesis. The following briefly describes the three types of transformations that comprise Glow.

**Actnorm.**  Kingma and Dhariwal (2018) proposed an activation normalization layer (Actnorm) as an alternative for batch normalization (Ioffe and Szegedy, 2015) to alleviate the challenges in model training. Similar to batch normalization, Actnorm performs an affine transformation of the activations using a scale and bias parameter per channel for 2D images, such that

$$y_{i,j} = s \odot x_{i,j} + b,$$

where both $x$ and $y$ are tensors of shape $[h \times w \times c]$ with spatial dimensions $(h, w)$ and channel dimension $c$.

**Invertible $1 \times 1$ convolution.**  To incorporate a permutation along the channel dimension, Glow includes a trainable invertible $1 \times 1$ convolution layer to generalize the permutation operation as:

$$y_{i,j} = W x_{i,j},$$

where $W$ is the weight matrix with shape $c \times c$.

**Affine Coupling Layers.**  Following Dinh et al. (2016), Glow includes affine coupling layers in its architecture of:

$$\begin{aligned} x_a, x_b &= \mathrm{split}(x) \\ y_a &= x_a \\ y_b &= \mathrm{s}(x_a) \odot x_b + \mathrm{b}(x_a) \\ y &= \mathrm{concat}(y_a, y_b), \end{aligned}$$

where $\mathrm{s}(x_a)$ and $\mathrm{b}(x_a)$ are outputs of two neural networks with $x_a$ as input. The $\mathrm{split}()$ and $\mathrm{concat}()$ functions perform operations along the channel dimension.

From this designed architecture of Glow, we see that interactions between spatial dimensions are incorporated only in the coupling layers. The coupling layer, however, is typically costly for memory resources, making it infeasible to stack a significant number of coupling layers into a single model, especially when processing high-resolution images. The main goal of this work is to design a new type of transformation that simultaneously models the dependencies in both the spatial and channel dimensions while maintaining a relatively small memory footprint to improve the capacity of the generative flow.

## 3  Masked Convolutional Generative Flows

In this section, we describe the architectural components of the *masked convolutional generative flow* (MACOW). First, we introduce the proposed flow transformation using masked convolutions in §3.1. Then, we present a fine-grained version of the multi-scale architecture adopted by previous generative flows (Dinh et al., 2016; Kingma and Dhariwal, 2018) in §3.2.

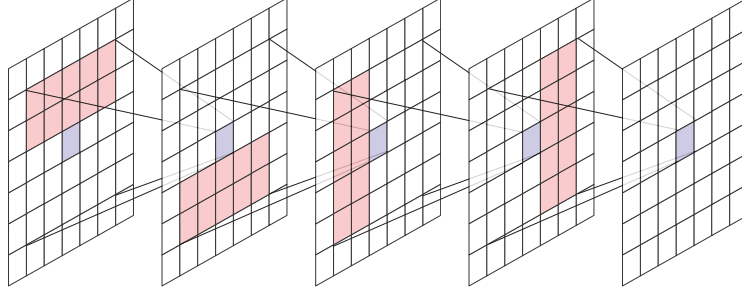

Figure 1: Visualization of the receptive field of four masked convolutions with rotational ordering.

### 3.1 Flow with Masked Convolutions

Applying autoregressive models to normalizing flows has been previously explored in studies (Kingma et al., 2016; Papamakarios et al., 2017), with idea of sequentially modeling the input random variables in an autoregressive order to ensure the model cannot read input variables behind the current one:

$$y_t = \mathrm{s}(x_{<t}) \odot x_t + \mathrm{b}(x_{<t}), \tag{4}$$

where $x_{<t}$ denotes the input variables in $x$ positioned ahead of $x_t$ in the autoregressive order. $\mathrm{s}()$ and $\mathrm{b}()$ are two autoregressive neural networks typically implemented using spatial masks (Germain et al., 2015; Oord et al., 2016).

Despite effectiveness in high-dimensional space, autoregressive flows suffer from two crucial problems: (1) The training procedure is unstable when stacking multiple layers to increase the flow capacities for complex distributions. (2) Inference and synthesis are inefficient, due to the non-parallelizable inverse function.

We propose to use masked convolutions to restrict the local connectivity in a small "masked" kernel to address these two problems. The two autoregressive neural networks, $\mathrm{s}()$ and $\mathrm{b}()$, are implemented with one-layer masked convolutional networks with small kernels (e.g. $2 \times 5$ in Figure 1) to ensure they only read contexts in a small neighborhood based on:

$$\mathrm{s}(x_{<t}) = \mathrm{s}(x_{t\star}), \quad \mathrm{b}(x_{<t}) = \mathrm{b}(x_{t\star}), \tag{5}$$

where $x_{t\star}$ denotes the input variables, restricted in a small kernel, on which $x_t$ depends. By using masks in rotational ordering and stacking multiple layers of flows, the model captures a large receptive field (see Figure 1), and models dependencies in both the spatial and channel dimensions.

**Efficient Synthesis.** As discussed above, synthesis from autoregressive flows is inefficient since the inverse must be computed by sequentially traversing through the autoregressive order. In the context of 2D images with shape $[h \times w \times c]$, the time complexity of synthesis is quadratic, i.e. $O(h \times w \times \mathrm{NN}(h, w, c))$, where $\mathrm{NN}(h, w, c)$ is the time of computing the outputs from the neural network $\mathrm{s}()$ and $\mathrm{b}()$ with input shape $[h \times w \times c]$. In our proposed flow with masked convolutions, computation of $x_{i,j}$ begins as soon as all $x_{t\star}$ are available, contrary to the autoregressive requirement that all $x_{<i,j}$ must have been already computed. Moreover, at each step we only need to feed a slice of the image (with shape $[h \times kw \times c]$ or $[kh \times w \times c]$ depending on the direction of the mask) into $\mathrm{s}()$ and $\mathrm{b}()$. Here $[kh \times kw \times c]$ is the shape of the kernel in the convolution. Thus, the time complexity reduces significantly from quadratic to linear, which is $O(h \times \mathrm{NN}(kh, w, c))$ or $O(w \times \mathrm{NN}(kw, h, c))$ for horizontal and vertical masks, respectively.

**Discussion** The previous work closely related to MACOW is the Emerging Convolutions proposed in Hoogeboom et al. (2019). There are two main differences. i) the pattern of the mask is different. Emerging Convolutions use "causal masks" (Oord et al., 2016) whose inverse function falls into a complete autoregressive transformation. In contrast, MACOW achieves significantly more efficient inference and sampling (§4.3), due to the carefully designed masks (Figure 1). ii) the Emerging Convolutional Flow, proposed as an alternative to the invertible $1 \times 1$ convolution in Glow, is basically a linear transformation with masked convolutions, which does not introduce "nonlinearity" to the random variables. MACOW, however, introduces such nonlinearity similar to the coupling layers.

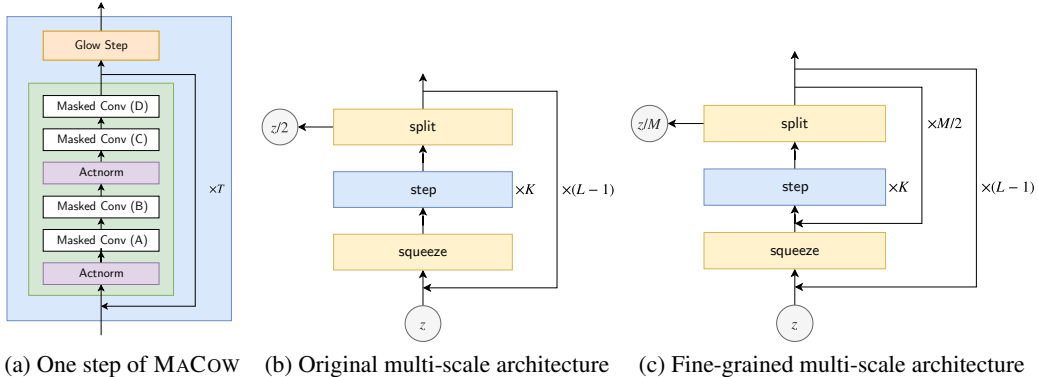

(a) One step of MACOW    (b) Original multi-scale architecture    (c) Fine-grained multi-scale architecture

Figure 2: The architecture of the proposed MACOW model, where each step (a) consists of $T$ units of *ActNorm* followed by two *masked convolutions* with rotational ordering, and a Glow step. This flow is combined with either an original multi-scale (b) or a fine-grained architecture (c).

## 3.2 Fine-grained Multi-Scale Architecture

Dinh et al. (2016) proposed a multi-scale architecture using a squeezing operation, which has been demonstrated to be helpful for training very deep flows. In the original multi-scale architecture, the model factors out half of the dimensions at each scale to reduce computational and memory costs. In this paper, inspired by the size upscaling in subscale ordering (Menick and Kalchbrenner, 2019) that generates an image as a sequence of sub-images with equal size, we propose a fine-grained multi-scale architecture to improve model performance further. In this fine-grained multi-scale architecture, each scale consists of $M/2$ blocks, and after each block, the model splits out $1/M$ dimensions of the input[1]. Figure 2 illustrates the graphical specification of the two architecture versions. Note that the fine-grained architecture reduces the number of parameters compared with the original architecture with the same number of flow layers. Experimental improvements demonstrate the effectiveness of the fine-grained multi-scale architecture (§4).

## 4    Experiments

We evaluate our MACOW model on both low-resolution and high-resolution datasets. For a step of MACOW, we use $T = 2$ masked convolution units, and the Glow step is the same as that described in Kingma and Dhariwal (2018) where an *ActNorm* is followed by an *Invertible* $1 \times 1$ *convolution*, which is followed by a *coupling layer*. Each coupling layer includes three convolution layers where the first and last convolutions are $3 \times 3$, while the center convolution is $1 \times 1$. For low-resolution images, we use affine coupling layers with 512 hidden channels, while for high-resolution images we use additive layers with 256 hidden channels to reduce memory cost. ELU (Clevert et al., 2015) is used as the activation function throughout the flow architecture. For variational dequantization, the dequantization noise distribution $q_\phi(u|x)$ is modeled with a conditional MACOW with shallow architecture. Additional details on architectures, results, and analysis of the conducted experiments are provided in Appendix B.

### 4.1    Low-Resolution Images

We begin our experiments with an evaluation of the density estimation performance of MACOW on two low-resolution image datasets that are commonly used to evaluate the deep generative models: CIFAR-10 with images of size $32 \times 32$ (Krizhevsky and Hinton, 2009) and the $64 \times 64$ downsampled version of ImageNet (Oord et al., 2016).

We perform experiments to dissect the effectiveness of each component of our MACOW model with ablation studies. The Org model utilizes the original multi-scale architecture, while the +fine-grained model augments the original one with the fine-grained multi-scale architecture proposed in §3.2. The

Table 1: Density estimation performance on CIFAR-10 $32 \times 32$ and ImageNet $64 \times 64$. Results are reported in *bits/dim*.

|  | Model | CIFAR-10 | ImageNet-64 |
|---|---|---|---|
| Autoregressive | IAF VAE (Kingma et al., 2016) | 3.11 | – |
|  | Parallel Multiscale (Reed et al., 2017) | – | 3.70 |
|  | PixelRNN (Oord et al., 2016) | 3.00 | 3.63 |
|  | Gated PixelCNN (van den Oord et al., 2016) | 3.03 | 3.57 |
|  | MAE (Ma et al., 2019) | 2.95 | – |
|  | PixelCNN++ (Salimans et al., 2017) | 2.92 | – |
|  | PixelSNAIL (Chen et al., 2017) | **2.85** | **3.52** |
|  | SPN (Menick and Kalchbrenner, 2019) | – | **3.52** |
| Flow-based | Real NVP (Dinh et al., 2016) | 3.49 | 3.98 |
|  | Glow (Kingma and Dhariwal, 2018) | 3.35 | 3.81 |
|  | Flow++: Unif (Ho et al., 2019) | 3.29 | – |
|  | Flow++: Var (Ho et al., 2019) | **3.09** | 3.69 |
|  | MACOW: Org | 3.31 | 3.78 |
|  | MACOW: +fine-grained | 3.28 | 3.75 |
|  | MACOW: +var | 3.16 | **3.69** |

Table 2: Negative log-likelihood scores for 5-bit LSUN and CelebA-HQ datasets in bits/dim.

| Model | CelebA-HQ | LSUN bedroom | tower | church |
|---|---|---|---|---|
| Glow (Kingma and Dhariwal, 2018) | 1.03 | 1.20 | – | – |
| SPN (Menick and Kalchbrenner, 2019) | **0.61** | – | – | – |
| MACOW: Unif | 0.95 | 1.16 | 1.22 | 1.36 |
| MACOW: Var | 0.67 | **0.98** | **1.02** | **1.09** |

+var model further implements the variational dequantization on the top of +fine-grained to replace the uniform dequantization (see Appendix A for details). For each ablation, we slightly adjust the number of steps in each level so that all the models have a similar number of parameters.

Table 1 provides the density estimation performance for different variations of our MACOW model along with the top-performing autoregressive models (first section) and flow-based generative models (second section). First, on both datasets, fine-grained models outperform Org ones, demonstrating the effectiveness of the fine-grained multi-scale architecture. Second, with the uniform dequantization, MACOW combined with the fine-grained multi-scale architecture significantly improves the performance over Glow on both datasets, and obtains slightly better results than Flow++ on CIFAR-10. In addition, with variational dequantization, MACOW achieves comparable result in bits/dim with Flow++ on ImageNet $64 \times 64$. On CIFAR-10, however, the performance of MaCow is around 0.07 bits/dim behind Flow++.

Compared with the state-of-the-art autoregressive generative models PixelSNAIL (Chen et al., 2017) and SPN (Menick and Kalchbrenner, 2019), the performance of MACOW is approximately 0.31 bits/dim worse on CIFAR-10 and 0.14 worse on ImageNet $64 \times 64$. Further improving the density estimation performance of MACOW on natural images is left to future work.

## 4.2 High-Resolution Images

We next demonstrate experimentally that our MACOW model is capable of high fidelity samples at high-resolution. Following Kingma and Dhariwal (2018), we choose the CelebA-HQ dataset (Karras et al., 2018), which consists of 30,000 high-resolution images from the CelebA dataset (Liu et al., 2015), and the LSUN (Yu et al., 2015) datasets including categories *bedroom*, *tower* and *church*. We train our models on 5-bit images with the fine-grained multi-scale architecture and both the uniform and variational dequantization. For each model, we adjust the number of steps in each level so that all the models have similar numbers of parameters with Glow for a fair comparison.

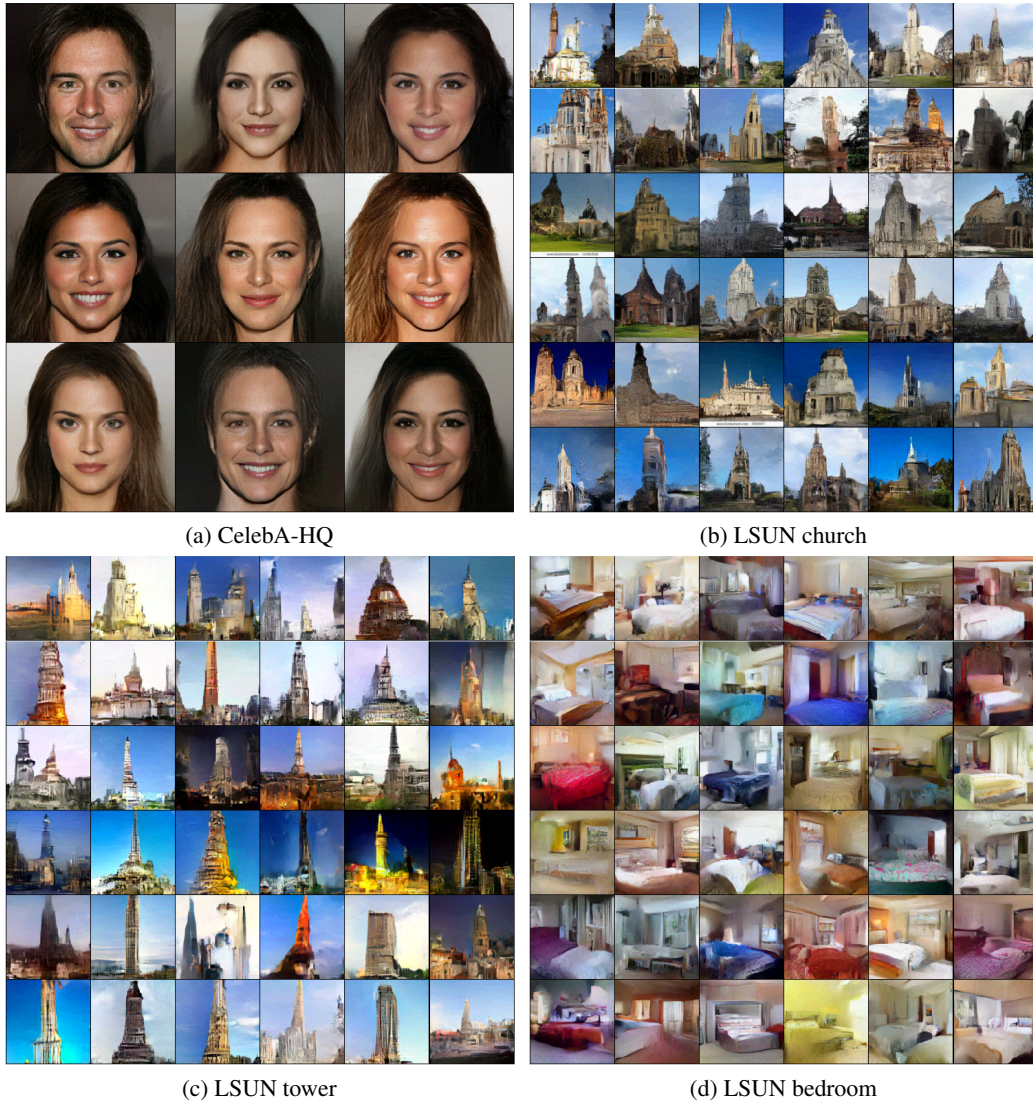

|     |     |
| --- | --- |
| (a) CelebA-HQ | (b) LSUN church |
| (c) LSUN tower | (d) LSUN bedroom |

Figure 3: (a) 5-bit $256 \times 256$ CelebA-HQ samples with temperature 0.7; (b)(c)(d) 5-bit $128 \times 128$ LSUN church, tower and bedroom samples, with temperature 0.9, respectively.

### 4.2.1 Density Estimation

Table 2 illustrates the negative log-likelihood scores in bits/dim of two versions of MACOW on the 5-bit $128 \times 128$ LSUN and $256 \times 256$ CelebA-HQ datasets. With uniform dequantization, MACOW improves the log-likelihood over Glow from 1.03 bits/dim to 0.95 bits/dim on CelebA-HQ dataset. Equipped with variational dequantization, MACOW obtains 0.67 bits/dim, which is 0.06 bits/dim behind the state-of-the-art autoregressive generative model SPN (Menick and Kalchbrenner, 2019) and significantly narrows the gap. On the LSUN datasets, MACOW with uniform dequantization outperforms Glow with 0.4 bits/dim on the bedroom category. With variational dequantization, the model achieves further improvements on all the three categories of LSUN datasets,

### 4.2.2 Image Generation

Consistent with previous work on likelihood-based generative models (Parmar et al., 2018; Kingma and Dhariwal, 2018), we found that sampling from a reduced-temperature model often results in higher-quality samples. Figure 3 showcases some random samples for 5-bit CelebA-HQ $256 \times 256$ with temperature 0.7 and LSUN $128 \times 128$ with temperature 0.9. The images are extremely high

Table 3: (a) Image synthesis speed on CIFAR10. Glow re-implemented in PyTorch is masked with
†. ‡ denotes results shown in Hoogeboom et al. (2019). (b) Image synthesis speed of MACOW on
datasets with different image sizes. The time is measured in milliseconds to sample a datapoint when
computed in mini-batchs with size 100.

<table>
<tr><td colspan="3" align="center">(a)</td><td colspan="3" align="center">(b)</td></tr>
<tr><td>CIFAR10</td><td>time (ms)</td><td>Slow-down</td><td>Dataset</td><td>image size</td><td>time (ms)</td></tr>
<tr><td>Glow‡</td><td>5</td><td>1.0</td><td>CIFAR10</td><td>$32 \times 32$</td><td>38.7</td></tr>
<tr><td>MAF ‡</td><td>3000</td><td>600.0</td><td>ImageNet</td><td>$64 \times 64$</td><td>104.7</td></tr>
<tr><td>Emerging‡</td><td>1800</td><td>360.0</td><td>LSUN</td><td>$128 \times 128$</td><td>267.9</td></tr>
<tr><td>Glow†</td><td>5.3</td><td>1.0</td><td>CelebA-HQ</td><td>$256 \times 256$</td><td>434.2</td></tr>
<tr><td>MACOW</td><td>38.7</td><td>7.3</td><td></td><td></td><td></td></tr>
</table>

quality for non-autoregressive likelihood models, despite that maximum likelihood is a principle that
values diversity over sample quality in a limited capacity setting (Theis et al., 2016). More samples
of images, including samples of low-resolution ones, are provided in Appendix C[2].

### 4.3 Comparison on Synthesis Speed

In this section, we compare the synthesis speed of MACOW at test time with that of Glow (Kingma
and Dhariwal, 2018), Masked Autoregressive Flows (MAF) (Papamakarios et al., 2017) and Emerging
Convolutions (Hoogeboom et al., 2019). Following Hoogeboom et al. (2019), we measure the time
to sample a datapoint when computed in mini-batchs with size 100. For fair comparison, we re-
implemented Glow using PyTorch (Paszke et al., 2017), and all experiments are conducted on a single
NVIDIA TITAN X GPU.

Table 3a shows the sampling speed of MACOW on CIFAR-10, together with that of the baselines.
MACOW is 7.3 times slower than Glow, much faster than Emerging Convolution and MAF, whose
factors are 360 and 600 respectively. The sampling speed of MACOW on datasets with different
image sizes is shown in Table 3b. We see that the time of synthesis increases approximately linearly
with the increase of image resolution.

## 5 Conclusion

In this paper, we propose a new type of generative flow, coined MACOW, which exploits masked
convolutional neural networks. By restricting the local dependencies in a small masked kernel,
MACOW boasts fast and stable training as well as efficient sampling. Experiments on both low-
and high-resolution benchmark datasets of images show the capability of MACOW on both density
estimation and high-fidelity generation, achieving state-of-the-art or comparable likelihood as well as
its superior quality of samples compared to previous top-performing models[3]

A potential direction for future work is to extend MACOW to other forms of data, in particular text,
on which no attempt (to the best of our knowledge) has been made to apply flow-based generative
models. Another exciting direction is to combine MACOW with variational inference to automatically
learn meaningful (low-dimensional) representations from raw data.

## Footnotes

[1]In our experiments, we set $M = 4$. Note that the original multi-scale architecture is a special case of the fine-grained version with $M = 2$.

[2]The reduced-temperature sampling is only applied to LSUN and CelebA-HQ 5-bits images, where MACOW

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
