[Supplementary Material]

# Appendix: MaCow: Masked Convolutional Generative Flow

## A   Dequantization

As described in §2, generative flows are defined on continuous random variables. Many real-world datasets, however, are recordings of discrete representations of signals, and fitting a continuous density model to discrete data produces a degenerate solution that places all probability mass on discrete datapoints (Uria et al., 2013; Ho et al., 2019). A common solution to this problem is "dequantization" that converts the discrete data distribution into a continuous one.

Specifically, in the context of natural images, each dimension (pixel) of the discrete data $x$ takes on values in $\{0, 1, \ldots, 255\}$. The dequatization process adds continuous random noise $u$ to $x$, resulting a continuous data point of:

$$y = x + u, \tag{1}$$

where $u \in [0, 1)^d$ is continuous random noise taking values from interval $[0, 1)$. By modeling the density of $Y \in \mathcal{Y}$ with $p_\theta(y)$, the distribution of $X$ is defined as:

$$P_\theta(x) = \int_{\mathcal{Y}} p_\theta(y)\, \mathrm{d}y = \int_{[0,1)^d} p_\theta(x + u)\, \mathrm{d}u. \tag{2}$$

By restricting the range of $u$ in $[0, 1)$, the mapping between $y$ and a pair of $x$ and $u$ is bijective. Thus, we have $p_\theta(y) = p_\theta(x + u) = p_\theta(x, u)$.

By introducing a *dequantization noise distribution* $q(u|x)$, the training objective in (1) can be re-written as:

$$
\begin{aligned}
\mathrm{E}_{P(X)}\Big[-\log P_\theta(X)\Big] &= \mathrm{E}_{P(X)}\left[-\log \int_{[0,1)^d} p_\theta(X, u)\, \mathrm{d}u\right] \\
&= \mathrm{E}_{P(X)}\left[\mathrm{E}_{q(u|X)}\left[-\log \frac{p_\theta(X, u)}{q(u|X)}\right] - \mathrm{KL}\big(q(u|X)\|p_\theta(u|X)\big)\right] \\
&\leq \mathrm{E}_{P(X)}\left[\mathrm{E}_{q(u|X)}\Big[-\log p_\theta(X, u)\Big] + \mathrm{E}_{q(u|X)}\Big[\log q(u|X)\Big]\right] \\
&= \mathrm{E}_{p(Y)}\Big[-\log p_\theta(Y)\Big] + \mathrm{E}_{P(X)}\mathrm{E}_{q(u|X)}\Big[\log q(u|X)\Big],
\end{aligned} \tag{3}
$$

where $p(y) = P(x)q(u|x)$ is the distribution of the dequantized variable $Y$ under the dequantization noise distribution $q(u|X)$.

**Uniform Dequantization.**   The most common dequantization method in prior work is uniform dequantization where the noise $u$ is sampled from the uniform distribution $\mathrm{Unif}(0, 1)$ such that

$$q(u|x) \sim \mathrm{Unif}(0, 1), \forall x \in \mathcal{X}.$$

From (3), we have

$$\mathrm{E}_{P(X)}\left[-\log P_\theta(X)\right] \leq \mathrm{E}_{p(Y)}\left[-\log p_\theta(Y)\right],$$

as $\log q(u|x) = 0, \forall x \in \mathcal{X}$.

**Variational Dequantization.**   As discussed in Ho et al. (2019), uniform dequantization directs $p_\theta(y)$ to assign uniform density to unit hypercubes $[0, 1)^d$, which is difficult for smooth distribution approximators. They proposed a parametric dequantization noise distribution $q_\phi(u|x)$ with a training objective to optimize the *evidence lower bound* (ELBO) provided in (3):

$$\min_{\theta, \phi} \mathrm{E}_{p_\phi(Y)}\left[-\log p_\theta(Y)\right] + \mathrm{E}_{P(X)}\mathrm{E}_{q_\phi(u|X)}\left[\log q_\phi(u|X)\right], \tag{4}$$

where $p_\phi(y) = P(x)q_\phi(u|x)$. In this paper, we implemented both these two dequantization methods for our MACOW, as is detailed in §4.

# B Experimental Details

## B.1 Model details

Table 4: Hyper-parameters for MACOW in our experiments.

| DataSet | Dequant | Batch Size | Levels | Depths per Level | # Param | # Param Glow |
|---|---|---|---|---|---|---|
| CIFAR-10 | Unif | 512 | 3 | $[[12, 12], [12, 12], 12]$ | 41.2M | 44.2M |
|  | Var | 512 | 3 | $[[12, 12], [12, 12], 12]$ | 43.5M |  |
| ImageNet | Unif | 160 | 4 | $[[16, 16], [16, 16], [12, 12], 12]$ | 117.2M | 111.6M |
|  | Var | 160 | 4 | $[[16, 16], [16, 16], [12, 12], 12]$ | 122.5M |  |
| LSUN | Unif | 160 | 5 | $[[32, 32], [32, 32], [16, 16], [12, 12], 6]$ | 166.6M | 198.1M |
|  | Var | 160 | 5 | $[[32, 32], [32, 32], [16, 16], [12, 12], 6]$ | 171.9M |  |
| CelebA-HQ | Unif | 40 | 6 | $[[24, 24], [16, 16], [16, 16], [8, 8], [4, 4], 2]$ | 171.9M | 170.8M |
|  | Var | 40 | 6 | $[[24, 24], [16, 16], [16, 16], [8, 8], [4, 4], 2]$ | 177.3M |  |

## B.2 Optimization

Parameter optimization is performed with the Adam optimizer (Kingma and Ba, 2014) with $\beta = (0.9, 0.999)$ and $\epsilon = 1e - 8$. Warmup training is applied to all the experiments: the learning rate linearly increases to for 500 updates to the initial learning rate $1e - 3$. Then we use exponential decay to decrease the learning rate with decay rate is $0.999997$.

# C  More samples from our experiments

Figure 4: Samples from 5-bit, $128 \times 128$ LSUN bedrooms.

Figure 5: Samples from 5-bit, 128×128 LSUN church.

Figure 6: Samples from 5-bit, 128×128 LSUN towers.

Figure 7: Synthetic celebrities sampled from 5-bit 256×256 CelebA-HQ.

Figure 8: Samples from 8-bit imagenet $64 \times 64$ with uniform dequantization

Figure 9: Samples from 8-bit imagenet $64\times64$ with variational dequantization