[Reviews · NeurIPS 2019]

Reviewer 1



Originality: This model is primarily composed of two main modifications to the original Glow model. The first modification is what makes it "masked". Instead of modeling the entire image as in glow, MaCow creates a semi-autoregressive model by allowing dependencies solely within a specific area using masked convolutions. This allows more efficient inference (effectively O(h) or O(w), rather than O(hw)) than a completely autoregressive model and higher quality modeling than a completely non-autoregressive model. Additionally, they tweak Glow to have more output latent channels in every scale that is being modeled, as is depicted in Figure 2c. Besides these two modifications, the model and setup greatly resembles that of the original Glow paper for image synthesis. Clarity: The paper is clear. However, the fine-grained architecture and the dequantization could be explained significantly more clearly. Significance: This work describes an intermediate between fully autoregressive and non-autoregressive flow models. Autoregressive models tend to be better at density estimation, so somewhat naturally this model achieves better results on density estimation.

Reviewer 2



UPDATE: Many thanks to the authors for the rebuttal, which clearly answers many of our questions. I have increased my score to 6 in response. I'm glad to see measurements of generation speed, and I think these will improve the paper. They experimentally confirm that generation speed scales linearly with the height (or width) of the image. My apologies for thinking that s() and b() are linear. Of course they can contain masked convolutions as well as nonlinearities. The following statement in the paper "The two autoregressive neural networks, s() and b(), are implemented with one-layer masked convolutional networks" gave me the impression that s() and b() are a single masked convolution. This misunderstanding demonstrates that the explanation of masked convolutions can be improved. That said, I hope that the authors agree that masked convolutions are a specific way of implementing autoregressive convolutions (which exist in various forms already since e.g. PixelCNN), rather than a new conceptual development, which I believe justifies my judgement of low originality. In general, I'd like to re-iterate that the paper could be improved if it focused on its core contribution (which is the masked convolution) and explained it more clearly, Currently, the paper goes through a lot of material (such as variational dequantization) which is already known and orthogonal to the contribution of the paper. I sincerely hope the authors will take that into account when revising the paper. Summary: The paper describes MaCow, a flow-based generative model for images. Like Glow, MaCow includes affine coupling layers, 1x1 convolutions, and actnorm layers. In addition to these, MaCow includes masked convolutional layers and a fine-grained variant of the multi-scale architecture of Glow. The model achieves state-of-the-art results in image generation for flow-based models, as measured by bits/dim on a test set. Originality: The two original elements of this paper are the masked convolutional layers and the fine-grained multi-scale architecture. However, I would consider both of them to be rather incremental contributions, as they are variants of already existing architectures and not significantly novel. In particular, the masked convolution is essentially an autoregressive layer, whose scale and shift functions are linear, share parameters, and have a restricted receptive field, whereas the fine-grained architecture is an almost trivial variation of the multi-scale architecture of Real NVP. There are other versions of invertible convolutions not discussed in the paper, for example: Hoogeboom et al, Emerging Convolutions for Generative Normalizing Flows, arXiv:1901.11137, January 2019. I think the paper would benefit from a discussion of how the masked convolutional layers differ from invertible convolutions such as the above. Quality: The model is tested in four image datasets, two of which are high-resolution, and achieves state-of-the-art results in terms of bits/dim in some cases. The experimental evaluation is done well; I'm impressed by the ablation experiments and the level of detail provided in the supplementary material. The ablation experiments clearly evaluate the separate improvements due to the masked convolutional layers, the fine-grained multi-scale architecture and the variational dequantization; in particular, we see that the masked convolutional layers and the fine-grained architecture yield small improvements, whereas the variational dequantization yields a larger improvement. It would have been good to include error bars though, so that we could confirm that the improvements are statistically significant (they probably are). My concern is whether the masked convolutional layers slow down image generation. The masked convolutional layers are essentially autoregressive layers with restricted receptive fields; as a result, instead of HxW passes they require H (or W) passes. For a 256x256 image, does that mean that the masked convolutional layers are 256 times more expensive to invert than to evaluate? If that's the case, then I don't see the usefulness of the masked convolutions; they provide little benefit for such a reduction in generation speed. In line 128, it is stated that "the training procedure is unstable when modeling an extended range of contexts and stacking multiple layers". This is surprising claim, and I'm not sure I believe it. My understanding is that the instability is entirely due to stacking multiple layers, and has nothing to do with the size of the context. I think either more supporting evidence is required, or the claim should be amended. Clarity: The paper is generally well written and easy to understand for someone familiar with the field. However, the writing is sloppy at times, and the paper could benefit from a revision with more attention to detail. Moreover, the explanation of masked convolutional layers in section 3.1 is rather compressed, and could be made clearer. Since masked convolutional layers are the main contribution of the paper, I think the paper would benefit significantly if section 3.1 were expanded and made clearer. Significance: Even though the paper achieves state-of-the-art results, it does so by incrementally varying already existing architectures. For this reason, I don't consider the contribution of this paper to be particularly significant.

Reviewer 3



Given the Glow and flow++ papers, it seems the biggest contribution of this work is in the masked convolution coupling layers, which on its own, improves upon Glow but falls behind flow++ for both uniform and variational dequantization. Given the paper's abstract, I would've expected performance to exceed flow++ but this seems to not be the case for either uniform or variational dequantization. I would've liked to see more comparable experiments with flow++ (by simply augmenting the flow++ architecture with masked convolutions since their code is publicly available) especially since MaCow has the exact same motivations (judging from the abstract) and also uses variational dequantization. I found the current comparisons a bit messy and difficult to understand. Given that Glow had to use 1 batch per GPU and gradient checkpointing in order to train on CelebA-HQ, can the authors comment on how MaCow compares? The paper also dedicates a page to discussing dequantization; however, it isn't clear to me how this is different from flow++'s ELBO. It seems the main equations are simply re-deriving this lower bound on the log density. I did not attribute this to be a contribution of the paper, but it could be that I simply did not understand the message here. Note that the temperature trick for sampling used in Glow is only applicable when the change in log density is constant wrt. the sample, so they only applied it for additive coupling layers. Can the authors verify whether their experiments satisfy this property? On another note, please be careful with link submissions and author identity. Many of the items on the reproducibility checklist seems wrong, e.g. how hyperparameters were chosen was not specified, the number of evaluations runs is not specified, and I did not see a single error bar or standard deviation in the paper? --- I thank the authors for providing wallclock time for sampling. I have one comment being that the authors should clarify in the paper how their section on variational dequantization is different from flow++'s exposition, or I would recommend moving it to a background section and instead expanding the sections on the properties of masked convolutions for a clearer narrative.

[Author Response · NeurIPS 2019]

Table 1: (a) Image synthesis speed on CIFAR10. Glow re-implemented in PyTorch is masked with †. ‡ denotes results shown in Hoogeboom et al. [2019]. (b) Image synthesis speed of MaCow on different datasets.

(a)

| CIFAR10 | time/sample (ms) | Slow-down |
|---|---|---|
| Glow‡ | 5 | 1.0 |
| MAF ‡ | 3000 | 600.0 |
| Emerging‡ | 1800 | 360.0 |
| Glow† | 5.3 | 1.0 |
| MaCow | 38.7 | 7.3 |

(b)

| Dataset | image size | time/sample (ms) |
|---|---|---|
| CIFAR10 | $32 \times 32$ | 38.7 |
| ImageNet-64 | $64 \times 64$ | 104.7 |
| LSUN-128 | $128 \times 128$ | 267.9 |
| CelebA-HQ-256 | $256 \times 256$ | 434.2 |

We thank for the valuable feedback. We address the questions below and will revise our paper accordingly.

[R1 & R2 & R3 Generation speed] We appreciate your suggestions on adding experiments on image generation speed
comparison and plan to add a section in experiments to discuss it in details. Table 1a and Table 1b provides the
preliminary results. Following Hoogeboom et al. [2019], we measure the time to sample a datapoint when computed in
mini-batchs with size 100. For fair comparison, we re-implemented Glow using PyTorch and show that the speed of
Glow† and Glow‡ is comparable. On CIFAR-10, MaCow is 7.3 times slower than Glow, much faster than Emerging
Convolution and MAF, whose factors are 360 and 600 respectively. The generation speed of MaCow on different
datasets is shown in Table 1b. We see that the time of generation increases linearly with the the image resolution.

[R2 Related work and Originality] Thanks for pointing out the related work. We will cite the Emerging Convolution
paper in our final version and discuss the detailed relation and difference with MaCow. There are two main differences
between MaCow and Emerging Convolution. i) the pattern of the mask is different. With the mask in MaCow (Figure
1 in the paper), MaCow achieves significantly more efficient inference and sampling (shown in Table 1a and 1b) by
reducing the complexity from $O(h \times w)$ to $O(h)$ or $O(w)$, without sacrificing the receptive field. ii) the Emerging
Convolutional Flow [Hoogeboom et al., 2019] is basically a linear transformation with masked convolutional kernels,
which does not introduce "nonlinearity" to the random variables. This flow is proposed as an alternative to the $1 \times 1$
conv flow in Glow. MaCow, in contrast, is able to introduce such nonlinearity similar to the coupling layer, and is
proposed to replace both the coupling layers and the $1 \times 1$ conv layers in Glow. We would like to point out that your
comment "the masked conv layers in MaCow are linear" is a misconception. Actually, regardless of the different
patterns of masks, Emerging Conv is a special (linear) case of MaCow, by specifying the $s(x_{<t})$ and $b(x_{<t})$ in Eq (4)
as linear functions.

[R2 & R3 Novelty and significance of improvements] First, we would like to argue that the proposed new pattern of
mask in MaCow is simple and effective, but not trivial or incremental. It produces a semi-autoregressive model, which
significantly reduces the inference time of MAF and obtains better density estimation performance.

Second, the improvements of MaCow on bits/dim, especially the contribution of fine-grained multi-scale architecture,
are not neglectable. Emerging Convolution [Hoogeboom et al., 2019] obtained 0.02 improvement on bits/dim by
increasing both the number of parameters and inference time. Our fine-grained architecture reduces the number of
parameters (compared with the original multi-scale architecture) and obtains 0.03 bit/dim improvements on both
CIFAR-10 and ImageNet-64. The overall improvements of MaCow over Glow, without Variational dequantization, are
0.07 on CIFAR-10, 0.06 on ImageNet-64, 0.04 on LSUN-bedroom and 0.08 on CelebA-HQ.

[R3 Flow++ v.s. MaCow] From results shown in Table 1 of our submission, in terms of density estimation, we could find
that on CIFAR-10 with uniform dequantization, MaCow (3.28) performs better than Flow++ (3.29) and on ImageNet-64,
with variational dequantization, MaCow (3.66) outperforms Flow++ (3.69). The only exception is on CIFAR-10 with
variational dequantization, Flow++ (3.09) achieve better performance than Macow (3.16). But we have to mention that
even with similar number of parameters, Flow++ is slower and consumes much more memory than Glow and MaCow,
preventing us from evaluating it on high-resolution images.

[R3 Sampling with temperature] The temperature trick is only applied to LSUN and CelebA-HQ 5-bits images, where
MaCow adopts additive coupling layers. For CIFAR-10 and ImageNet 8-bits images, we sample with temperature 1.0.

[R1 non-numerical measures of modifications] We appreciate your suggestion on evaluating the difference between Ma-
Cow w./w.o. these modications in non-numerical ways. From the image samples w./w.o the variational dequantization,
we have not observed significant difference. We will consider some other non-numerical metrics.

[R2 & R3 Error bar] We have performed experiments on CIFAR-10 with multiple runs (more than 3) with different
random seeds and the standard deviation is less than 0.005. On other datasets, same with Glow and Flow++, we only
performed single run on each dataset due to the limits of computational resources, unfortunately.

[R2 statement of stability] We appreciate your comment about our irrigorous statement of stability in line 128. We will
amend this claim in the final version.

[R1 & R2 & R3 Paper writing] We thank for all your suggestions on revising the paper to improve the writing. We will
elaborate the Masked Convolution and Fine-grained architecture sections to explain them more clearly.

## References

Emiel Hoogeboom, Rianne V. Berg, and Max Welling. Emerging convolutions for generative normalizing flows. In *ICML*, 2019.


[Meta-Review · NeurIPS 2019]

The paper proposes two ways of improving flow-based models of images such as Glow: 1. introducing invertible (locally) autoregressive layers implemented using masked convolutions and 2. making the multi-scale architecture more fine-grained by "factoring" out the variables in stages rather than in one go. While the reviewers found both of these contributions quite incremental, the experimental section is quite strong, with informative ablation studies and state-of-the-art results. The paper really needs substantial reworking however, as the primary contributions are not described with sufficient clarity and detail, and too much space is used for explaining data dequantization, which is not novel and could be covered much more concisely.